# Free energy perturbations in enzyme kinetic models reveal cryptic epistasis

Karol Buda *, Nobuhiko Tokuriki *

Michael Smith Laboratories, University of British Columbia, Vancouver, British Columbia, Canada

* karol.buda@msl.ubc.ca (KB); tokuriki@msl.ubc.ca (NT)

## Abstract

Epistasis—the context-dependence of mutational effects—is a key driver of protein evolution, influencing adaptive pathways and functional diversity. While specific epistasis arises from direct physical interactions between mutations, non-specific epistasis emerges when a non-linear mapping links a protein's biophysical properties to its function. Enzyme kinetic parameters map directly to free energies, enabling researchers to connect epistasis in these parameters to an enzyme's structural features. Here, we show that this approach is incorrect: enzyme catalytic parameters like $k_{cat}$ and $K_M$ inherently exhibit non-specific epistasis due to the multi-state nature of the catalytic cycle. Using enzyme catalytic cycle models, parameterized by free energies of ground and transition states, we simulated 1000 "mutations" or perturbations to the sub-state free energies within the kinetic ensemble. We then combined these mutations, creating one million double mutants with strictly additive free energy effects. Despite the absence of explicit mutational interactions, we observed substantial epistasis in catalytic parameters; its prevalence and complexity increasing with the number of kinetic states in the mechanism. We derived analytical conditions for the emergence of this form of epistasis in a simple kinetic model, demonstrating that non-specific epistasis depends on the relative values of key microscopic rate constants. Finally, we validated our framework by reanalyzing kinetic data for double mutants in *Bacillus cereus* β-lactamase I and found that reported specific epistasis in catalytic efficiency was substantially stronger than previously inferred, altering mechanistic interpretations. Our results identify an intrinsic, previously unknown source of epistasis that can distort both the magnitude and sign of mutational effects in enzyme kinetics. We provide theoretical and computational tools for recognizing and correcting for this form of non-specific epistasis, enabling accurate mechanistic inference from kinetic data and improving our understanding of the links between epistasis, structure-function relationships, enzyme evolution, and protein design.

**Data availability statement:** We implemented the epistasis simulations using R (v4.5.0) supplemented with the libraries tidyverse (v2.0.0) and progress (v1.2.3). All simulations and software are available in the Supporting information (S1 and S3 Files) as well as on GitHub (https://github.com/karolbuda/kinetic-epistasis-simulations). All mutant data are provided at doi:10.5281/zenodo.17041247.

**Funding:** K.B. and N.T. are supported by the Natural Sciences and Engineering Research Council of Canada (NSERC; https://www.nserc-crsng.gc.ca/index_eng.asp)/Discovery Grants Program (RGPIN-2023-05135) and the Human Frontier Science Program (HFSP; https://www.hfsp.org/) Research Grant (RGP0054/2020). The funders had no role in study design, data collection and analysis, decision to publish, or preparation of the manuscript.

**Competing interests:** The authors have declared that no competing interests exist.

## Author summary

Enzymes help organisms convert reactants to products through a series of steps; each associated with an energy that dictates how well the enzyme catalyzes a reaction. Enzymes evolve to become more efficient, or catalyze new reactions, through mutations that change the free energies of these steps. Sometimes, the effect of one mutation depends on the presence of another, a phenomenon called epistasis. Epistasis is typically studied by measuring the effect of mutations on standard enzyme parameters under the assumption that changes in these values reflect structural interactions between mutations in the protein. Our study shows that this assumption is misleading. Even when mutations act independently on the energies of steps in an enzyme's reaction, when combined they can create epistasis. This phenomenon arises from the complex, non-linear relationships between the parameters that define each step in the enzyme reaction and the measurements we obtain during experiments probing enzyme functions. Using computational simulations, we mathematically derive the necessary conditions for this form of epistasis, demonstrate that epistasis increases in prevalence as the enzyme reaction becomes more complex, and apply our model to published experimental data. Our findings urge that researchers should account for these effects before drawing structural conclusions from epistasis.

## Introduction

Enzyme evolution is underpinned by the accumulation of adaptive, neo-functionalizing mutations. The effect of a mutation is dictated by the enzyme's sequence; beneficial mutations may be contingent on specific amino acid interactions or inaccessible due to certain incompatibilities within the sequence [1–9]. This mutational context-dependence, known as epistasis, dictates the accessibility of evolutionary paths that an enzyme can take, and is a major driver of sequence diversity in nature [10]. Epistasis can dramatically change the topology of an enzyme's fitness landscape and distort our ability to accurately assess the functional effect of a mutation across different enzymes [11–13]. Thus, our ability to predict and understand evolutionary outcomes requires a robust understanding of epistasis.

Epistasis is generally categorized using three criteria: its sign contribution relative to the effects of the individual mutations, the degree of deviation from the expected outcome relative to a wild-type (wt) reference state, and its source. When single mutational effects depart from their expected behaviour, the resulting double mutant's function will either be greater than expected (positive epistasis) or lower than expected (negative epistasis). Upon categorizing the sign contribution of the single mutational effects, epistasis can simply amplify or diminish the mutational effects (magnitude epistasis), entirely change their sign contribution, *e.g.*, from beneficial in one background to deleterious in another (sign epistasis), or, in the most extreme

case, change the sign contribution of both interacting mutations, *i.e.,* cause two positive mutations to elicit negative epista-sis relative to the wt or *vice versa* (reciprocal sign epistasis) [14,15]. Finally, the source of epistasis is broadly categorized into two main types: specific and non-specific [16]. Specific epistasis (also called idiosyncratic epistasis [12]) arises from specific physical interactions between mutations—these may be proximal or relayed through an enzyme's intramolecular network [17]. Functionally relevant interactions expose the intricate connections between amino acids within enzymes and help link enzyme structures to their functions [18–21]. A deep understanding of specific epistasis offers insight into the enzyme's mechanism and provides both limitations and opportunities for changes in the enzyme's sequence to promote existing chemical reactions or establish novel ones. However, extracting specific epistasis from functional measurements is difficult, as it is frequently distorted by non-specific epistasis.

At its core, non-specific epistasis (also referred to as global epistasis [12,22]) arises when mutational combinations appear epistatic due to some non-linear relationship between their effects on an enzyme's biophysical property and the enzyme's function [16,23]. In other words, non-specific epistasis arises from mutations whose free energy changes combine in an additive manner, but map non-linearly with respect to some functional parameter (in contrast to specific epistasis, where free energy changes from mutations themselves combine non-additively). The complexity of the non-specific epistasis is proportional to the complexity of the function that translates an enzyme's biophysical parameters to the measurable, functional readout, such as enzyme catalytic activity or whole organismal fitness. The simplest example is threshold epistasis [24,25], where the relationship between protein stability and expression (which, itself, is assumed to be proportional to function) is well defined by the Boltzmann distribution—a sigmoid function relating the folded proportion of a protein and the change in free energy ($\Delta G$) between folded and unfolded states. Changes in free energy, an additive trait, can result in non-additive changes to the folded fraction and thus, the observable function. Indeed, deep mutational scanning (DMS) experiments have unveiled these global sigmoidal relationships between predicted and observed func-tions of mutations [26–30]. Thus, a conventional practice is to estimate non-specific epistasis using sigmoidal or other similar threshold models that enact a non-linear transformation of the function to the linear free energy scale [31,32]. However, non-specific epistasis can be underpinned by more complex relationships. For example, if proteins sample more than two functionally relevant sub-states, the relationship between all sub-state $\Delta G$s and the protein's function becomes obscured. This was conceptually elucidated by Morrison *et al.* (2021) in the form of ensemble epistasis [33]. They demon-strated that, in the presence of more than two protein sub-states, protein function can be linked to a Boltzmann-weighted average of its conformational sub-states. When mutations that additively alter the free energies of each sub-state are combined, they can result in a non-additive change to the weighted average of the sub-states, resulting in complex pat-terns of epistasis, including sign epistasis [33]. These patterns cannot be captured by a threshold model, as the protein's phenotype is an aggregate of several non-linear functions.

Enzyme kinetics may be another example where non-specific epistasis can arise from the ensemble of sub-states. The enzyme catalytic cycle must go through several energetic states, *e.g.*, the enzyme-substrate complex and the enzyme-transition state (TS) complex. The relationship between the energetic level of each state and the commonly measured kinetic parameters ($k_{cat}$, $K_M$, and catalytic efficiency) can be multivariate and scales in complexity with the enzymatic mechanism. It has been empirically demonstrated that, in the absence of specific epistasis, mutational effects of sub-state free energies combine additively [34]. However, mutations can differentially affect the relative free energies of these sub-states, and thus, multiple mutational effects can further obscure the relationship between the sub-state energies and the measured kinetic parameters. Despite the well-established functions that connect sub-state energy to measured kinetic parameters, potential sources of non-specific epistasis arising from these non-linear relationships have not been explored. In this study, we aimed to uncover an untapped source of non-specific epistasis in enzymes. By assigning free energies to ground and transition states within an *in silico* catalytic cycle, and simulating mutations that perturb these free energies, we investigated whether additive free-energy changes between mutational combinations may create non-specific epistasis and provide a theoretical rationale for our observations. We also explored how increasing complexity in the catalytic cycle results in more non-specific

epistasis. Finally, we applied our model to experimental data and extracted non-specific epistasis in the catalytic cycle from a β-lactamase double mutant. Our work demonstrates that catalytic parameters appear to be inherently epistatic and cautions researchers from making direct structural claims using epistasis obtained from enzyme kinetics measurements.

## Results

### Simple kinetic model reveals emergent epistasis

To determine whether mutations with additive effects on energetic terms may create non-specific epistasis, we established a model with no explicit interactions where a reaction coordinate is based on a hypothetical enzyme reaction governed by the classical Michaelis-Menten enzyme mechanism (**Fig 1a**) [35]. The mechanism is defined by the following rate constants: a reversible binding step defined by rate constants $k_1$ and $k_{-1}$, as well as an irreversible chemical step of product formation, $k_2$, hereafter referred to as "microscopic rate constants." To obtain the microscopic rate constants of this hypothetical enzyme reaction, we assigned relative Gibbs free energies ($G$), in kcal mol$^{-1}$, to each state in the reaction coordinate: the ground states, *i.e.*, the enzyme and substrate (E + S) and the enzyme-substrate complex (ES), as well as the transition states, *i.e.*, the binding transition state (E + S)$^{‡}$ and the chemical transition state (ES)$^{‡}$ (**Fig 1a**). We then calculated each rate constant using the Arrhenius equation, employing transition state theory for the approximation of each pre-exponential factor (see Methods). Finally, we calculated the kinetic parameters: the substrate binding dissociation constant ($K_D$), turnover number ($k_{cat}$), the Michaelis-Menten constant ($K_M$), and catalytic efficiency ($k_{cat}/K_M$), as defined by the enzyme mechanism using the simulated microscopic rate constants. The values for relative free energies (**Fig 1a**) were chosen arbitrarily, with the requirement that the extrapolated $k_{cat}$, $K_M$, and $k_{cat}/K_M$ from the computed microscopic rate constants (**Table 1**) were within the range of experimentally measured, median kinetic parameters for natural enzymes outlined in studies by Bar-Even *et al.* (2011) [36] and Copley *et al.* (2022) [37].

Next, we simulated 1000 in silico "mutations" which are denoted by a perturbation to the free energies of each state within the reaction coordinate. A mutation perturbed each of the five sub-state's $G$ in the ensemble, both ground and transition states,

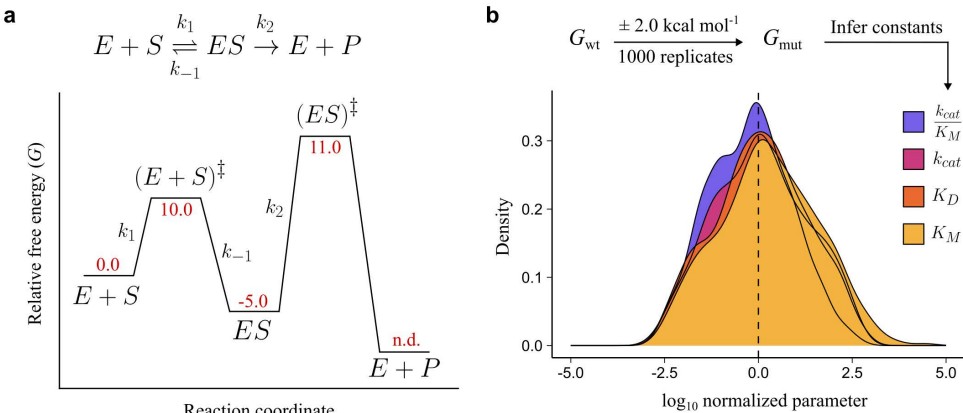

**Fig 1. Inferred enzyme constants across 1000 *in silico* mutations are log-normally distributed. a,** Overview of the mechanism and reaction coordinate of each state in the enzyme kinetic cycle. Red values represent Gibbs free energy for each state. E + P free energy was not defined (n.d.) as it is not required in rate constant calculation (see Methods). **b,** Distribution of inferred constants after log-transformation and normalization to the wt value.

**Table 1. Rate constants selected for the WT *in silico* state.**

| $k_{on}$ (M$^{-1}$ s$^{-1}$) | $k_{off}$ (s$^{-1}$) | $K_D$ (µM) | $k_{cat}$ (s$^{-1}$) | $K_M$ (µM) | $k_{cat}/K_M$ (M$^{-1}$ s$^{-1}$) |
|---|---|---|---|---|---|
| 2.88 x 10$^5$ | 62.1 | 215 | 11.5 | 255 | 4.50 x 10$^4$ |

by a random value between –2 and 2 kcal mol⁻¹, sampled from a uniform distribution (**Fig 1a**). We allowed for a single mutation to perturb multiple free energy states as this has, empirically, been demonstrated to be the norm across several enzymes [38–42]. The new sub-state *Gs* were recorded for each of the 1000 mutants and used to calculate new rate constants and kinetic parameters for the mutants. As expected, we found that the distributions of the mutants' kinetic parameters were log-normally distributed (**Fig 1b**). We then extrapolated a set of double mutants by summing the wt *G* to the two mutants' $\Delta G$ values for each state, resulting in $10^6$ (1000 x 1000) double mutants; rate constants and kinetic parameters were calculated for each mutant. We also computed the double mutants' predicted values for each individual rate constant and kinetic parameter by multiplying the wt value with the corresponding fold-changes in each constituent single mutant–a commonly employed null model that assumes additive mutational effects (see Methods). We employ the null model as our baseline, as, in the absence of epistasis, the assumption that free energy changes in double mutants should be equal to the sum of free energy changes in the single mutants–the central assumption of the null model–has been demonstrated empirically across several enzymes and other proteins [28,29,34,43]. All mutant data are made available in a public repository [44].

As we did not explicitly introduce mutational interactions into our model, we aimed to probe whether differences in the double mutants' predicted rate constants *versus* observed rate constants revealed non-specific epistasis. To account for the presence of noise in experimentally obtained rate constant measurements, we applied a 1.5-fold significance threshold for epistasis detection, which we have used previously [11]. We found that of the $10^6$ *in silico* variants, 39.4% showed significant epistasis in $k_{cat}/K_M$ (**Fig 2a**), with 28.6% at 2-fold, 7.1% at 5-fold, and 2.1% at 10-fold significance thresholds. For the 1.5-fold threshold, the majority of the epistasis was magnitude epistasis (33.0%), with some mutants exhibiting sign (5.6%) and reciprocal sign (0.8%) epistasis (**Fig 2a**). Among all observed epistasis, both positive and negative epistasis appeared: 59.9% (117,680/196,355) positive versus 40.1% (78,675/196,355) negative. We observed exactly inverse positive-negative ratios for epistasis in the $K_M$ (**S1 File**), but similar magnitude and sign distribution (**Fig 2b and S1 File**). However, we did not see any epistasis in $k_{cat}$ (**Fig 2c**) or $K_D$ (**Fig 2d**). The statistics for all kinetic parameters can be found in **S1 File**.

## Mechanisms for non-specific epistasis in a simple kinetic ensemble

The emergent epistasis from free energy perturbations in enzyme kinetics cannot be easily captured by a non-linear transformation (**Fig 2a**) such as a sigmoidal model or other threshold model, as is often the case for non-specific epistasis. These observations suggest that mutational behaviors in kinetic parameters are more complex compared to other biophysical properties. Thus, we conducted a theoretical exploration by dissecting how epistasis arises in $K_M$ (**Fig 2b**). We define epistasis in $K_M$ as deviation from a multiplicative null model, thus:

$$\varepsilon = \frac{\frac{K_M^{1,2}}{K_M^{wt}}}{\frac{K_M^1}{K_M^{wt}}\frac{K_M^2}{K_M^{wt}}}$$

(1)

And since $K_M$ is defined by the microscopic rate constants $k_{-1}$, $k_1$, and $k_2$, a single mutational effect on $K_M$ is simply the fold-change that a mutation elicits on each of the rate constants:

$$K_M^i = \frac{\alpha_i k_{-1} + \beta_i k_2}{\gamma_i k_1}$$

(2)

Where *i* is the relative mutational number (mutation one or mutation two), $\alpha$ is the mutation's fold-change on $k_{-1}$, $\beta$ is the mutation's fold-change on $k_2$, and $\gamma$ is the mutation's fold-change on $k_1$. Thus, the double mutational effect is:

$$K_M^{1,2} = \frac{\alpha_1 \alpha_2 k_{-1} + \beta_1 \beta_2 k_2}{\gamma_1 \gamma_2 k_1}$$

(3)

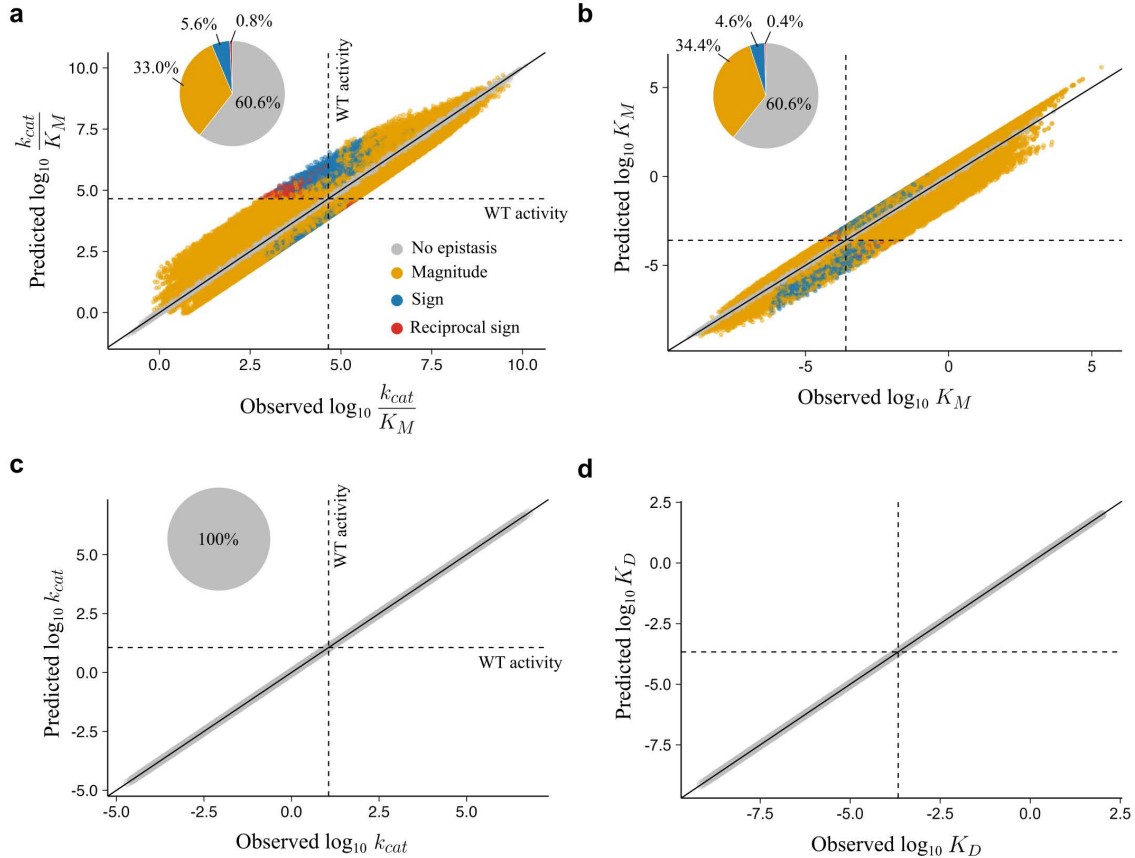

**Fig 2. Non-specific epistasis in catalytic efficiency arises from the null model.** Correlation of observed and predicted mutational effects with listed proportions of magnitude, sign, and reciprocal sign epistasis for the **a,** catalytic efficiency ($k_{cat}/K_M$) **b,** turnover number ($k_{cat}$), **c,** Michaelis-Menten constant ($K_M$) and **d,** enzyme-substrate dissociation constant $K_D$.

Upon substituting Eq. 1 with Eqs. 2, 3, and 11 (see Methods) and setting epistasis to zero (see **S2 File** for details), we found that:

$$(\alpha_1 - \beta_1)(\alpha_2 - \beta_2) = 0 \qquad (4)$$

In other words, non-specific epistasis in the simple kinetic ensemble only arises when each of the two mutations has a differential effect on $k_{-1}$ ($\alpha$) and effect on $k_2$ ($\beta$). Using a similar approach, we expand on the theoretical rationale for the absence of epistasis in $k_{cat}$ and $K_D$ in **S2 File**.

  Using our starting wt rate constants (**Table 1**), we computed the expected magnitude of epistasis in $K_M$ for mutational effects on $k_{-1}$ (*i.e.,* $\alpha_1$ for mutation one and $\alpha_2$ for mutation two) and $k_2$ (*i.e.,* $\beta_1$ for mutation one and $\beta_2$ for mutation two). We found that negative epistasis arose when mutations had opposite effects to each other, *e.g.,* one mutation increased $\alpha_i$ relative to $\beta_i$ while the other mutation decreased the ratio or *vice versa* (quadrants B and D in **Fig 3a**). Positive epistasis, on the other hand, arose only when both mutations decreased $\alpha_i$ relative to $\beta_i$ (quadrant A in **Fig 3a**), but not when both mutations increased the ratio, in which case no significant epistasis was observed (quadrant C in **Fig 3a**). We provide examples of how mutations can perturb sub-state free energies to produce double mutants that lie across different quadrants (**Fig 3b**). We also found that our simulated dataset of randomly sampled mutational combinations fit these

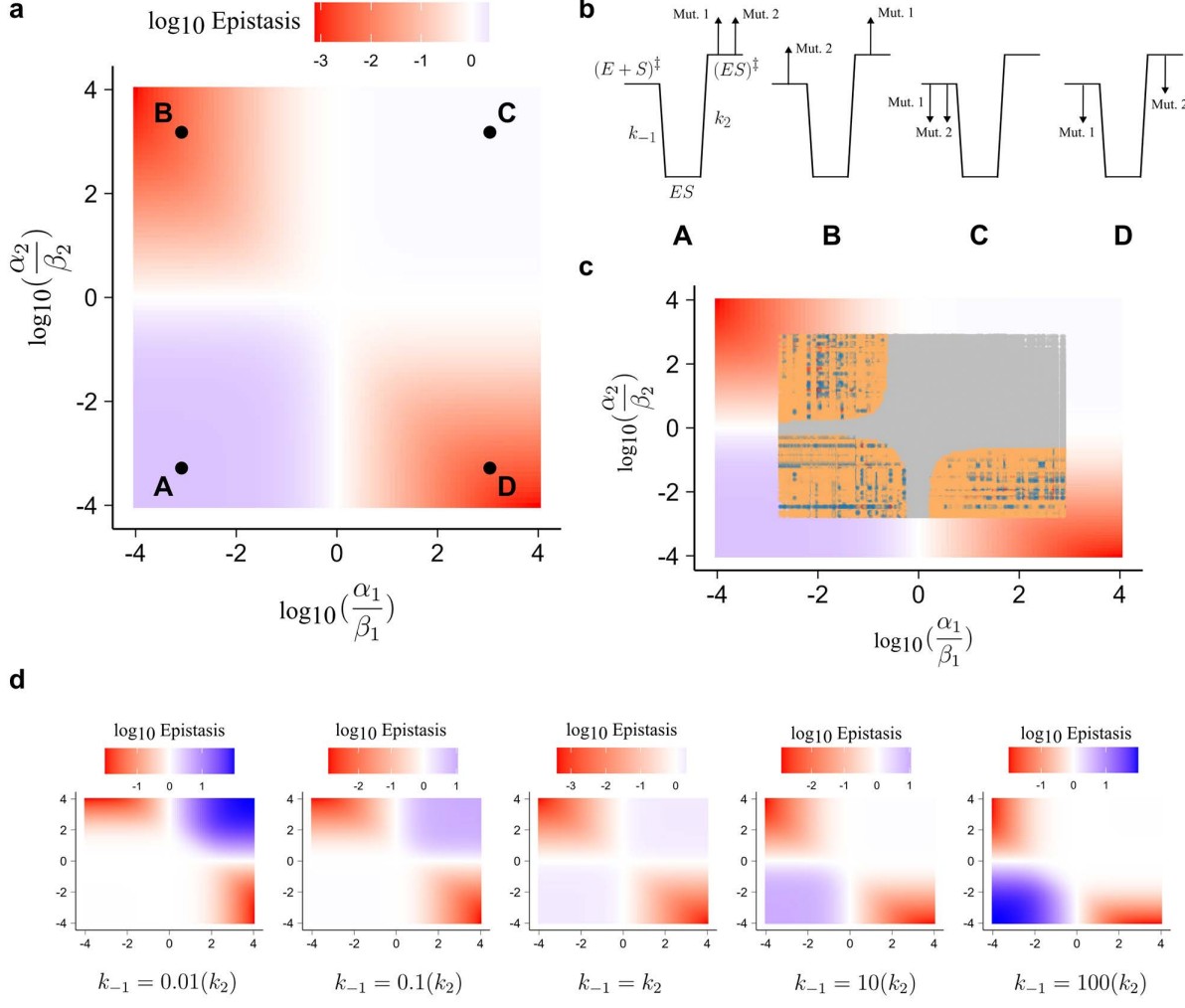

**Fig 3. Mutational effects on $k_{-1}$ and $k_2$ are sufficient to account for non-specific epistasis in the kinetic ensemble. a,** The observed non-specific epistasis as a function of ratios of mutations 1 effect on $k_{-1}$ ($\alpha_1$) and $k_2$ ($\beta_1$) versus mutations 2 effect on $k_{-1}$ ($\alpha_2$) and $k_2$ ($\beta_2$). Four extreme examples (A–D) are shown and explored in **b**. **b,** Four non-exhaustive examples of mutational effects that could lead to the observed patterns of epistasis in **a**. **c,** Distribution of simulated data from **Fig 2b** overlaid on the relationship shown in **a**. **d,** The change in observed non-specific epistasis with variation in starting values of $k_{-1}$ in the wt enzyme.

observations, as expected, and variants with magnitude, sign, and reciprocal sign epistasis were found in quadrants A, B, and D in **Fig 3c** (with quadrant C populated with variants that show no significant epistasis; **Fig 3c**). Moreover, we found that the relationship between the $\alpha_i/\beta_i$ and positive epistasis is sensitive to the starting value of $k_{-1}$ (**Fig 3d**). When $k_{-1} < k_2$, positive epistasis was only observed when both mutations increased the $\alpha_i/\beta_i$. When $k_{-1} = k_2$, no positive epistasis was observed. When $k_{-1} > k_2$ (equivalent to our simulated conditions), positive epistasis was only observed when both mutations decreased the $\alpha_i/\beta_i$. Furthermore, the strength of positive epistasis increased as $k_{-1}$ became much greater than or much less than $k_2$ (**Fig 3d**). Thus, the relative values of the key rate constants, as well as the degree to which mutations modulate them, are sufficient parameters for prediction and understanding of non-specific epistasis in kinetic parameters for simple enzyme mechanisms.

## Increasing kinetic model complexity amplifies epistasis

If epistasis can arise in a simple kinetic model and appears in kinetic parameters that are defined by the sum of multiple microscopic rate constants (**S2 File**), how does increasing the reaction mechanism complexity influence non-specific epistasis? To address this question, we established a reaction coordinate based on a more complex mechanism with a reversible chemical step, as well as an irreversible product release step (**Fig 4a**). The established reaction coordinate was rate-limited by the chemical step. We retained relative energies for states from the previous model and introduced energies for the new states. We note that these changes lowered the values for $k_{cat}$ and $K_M$ such that they deviate from median reported values across enzymes [36,37], although $k_{cat}/K_M$ was almost equivalent (**Table 2**). As with the previous model, we simulated 1000 mutations and monitored the emergent kinetic parameter distributions–these were also log-normally distributed (**Fig 4b**), albeit we found that the shape of the distributions, particularly that of the $K_M$, was broader than the simple model, likely owing to more complex equations underlying the kinetic parameters (see Methods).

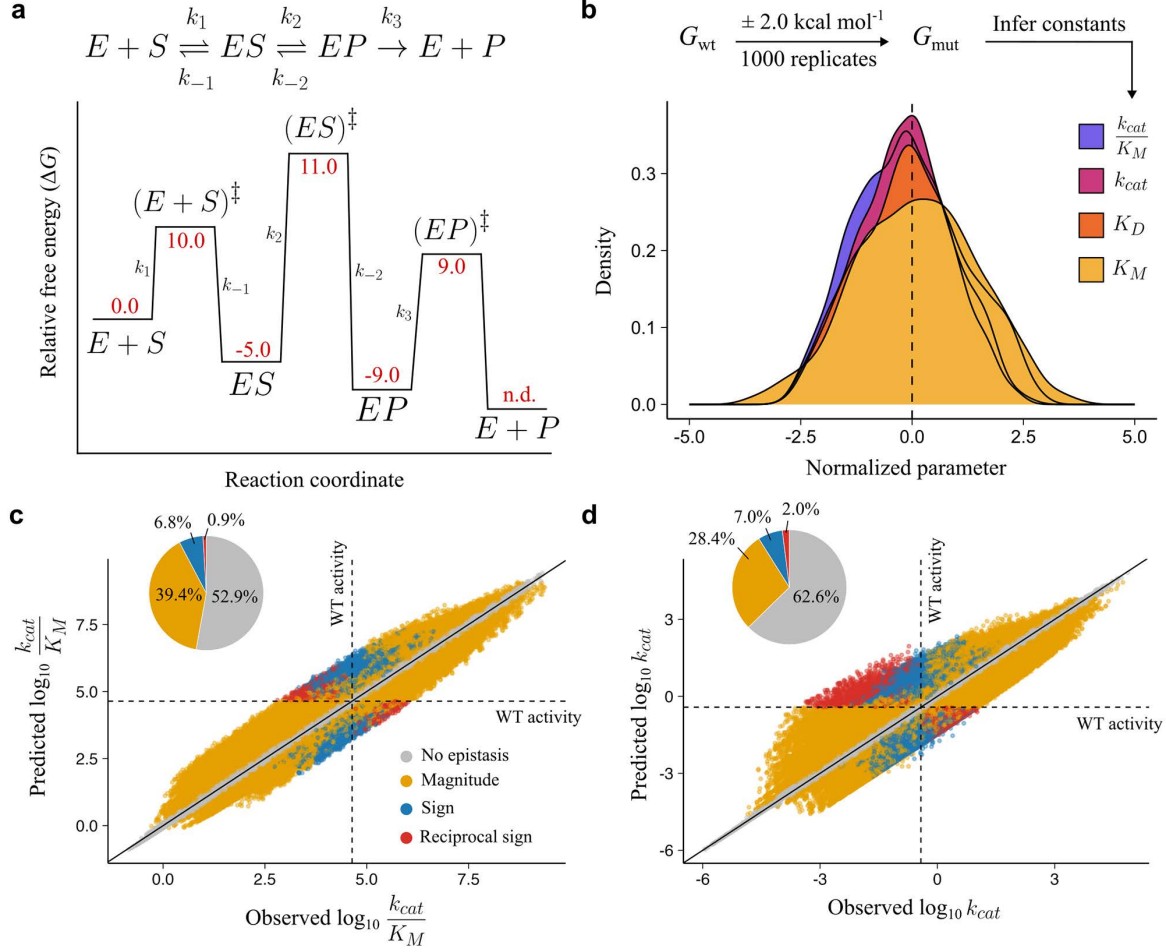

**Fig 4. Non-specific epistasis in catalytic efficiency increases with a more complex mechanism. a,** Overview of the mechanism and reaction coordinate of each state in the enzyme kinetic cycle. Red values represent Gibbs free energy for each state. E + P free energy was not defined (n.d.) as it is not required in rate constant calculation (see Methods). **b,** Distribution of inferred constants after log-transformation and normalization to the wt value. **c, d,** Correlation of observed and predicted mutational effects with listed proportions of magnitude, sign, and reciprocal sign epistasis for **c,** catalytic efficiency ($k_{cat}/K_M$) and **d,** $k_{cat}$.

**Table 2. Rate constants selected for the WT _in silico_ state in the complex model.**

| $k_{on}$ (M$^{-1}$ s$^{-1}$) | $k_{off}$ (s$^{-1}$) | $K_D$ (µM) | $k_{cat}$ (s$^{-1}$) | $K_M$ (µM) | $k_{cat}/K_M$ (M$^{-1}$ s$^{-1}$) |
|---|---|---|---|---|---|
| 2.88 x 10$^5$ | 62.1 | 215 | 0.38 | 8.6 | 4.37 x 10$^4$ |

As previously, $K_D$ did not show non-specific epistasis (**S3 File**). In addition to epistasis observed for $K_M$ (**S3 File**) and $k_{cat}/K_M$, we also observed non-specific epistasis in $k_{cat}$. The proportion of epistasis in $k_{cat}/K_M$ grew from 39.4% in the simple model to 47.1% (**Fig 4c**). The increase in epistasis was found across all forms: magnitude epistasis (39.4%), sign (6.8%) and reciprocal sign (0.9%) epistasis (**Fig 4c**). Like in the simple model, most of the significant epistasis was positive: 60.9% (143,060/234,976) positive versus 39.1% (91,916/234,976) negative. The epistasis in $k_{cat}$ was lower (37.4%), though the proportion of sign (7.0%) and reciprocal sign (2.0%) was greater than that of $k_{cat}/K_M$ in both models (**Fig 4d**). Furthermore, $k_{cat}$ epistasis appeared less skewed to positive effects than $k_{cat}/K_M$: 51.5% (95,842/186,050) positive vs 48.5% (90,208/186,050) negative. The statistics for all kinetic parameters can be found in the **S3 File**. All mutant data are made available in a public repository [44].

As expected, and outlined in the **S2 File**, unlike the simple model, we observed epistasis in $k_{cat}$ because it included a summation in its functional mapping to its constituent microscopic rate constants (eq. 4.1 in **Methods**). The equation for $K_D$, however, does not change with increasing complexity of the enzyme mechanism and therefore remains non-epistatic.

## Identification of epistasis from the catalytic cycle of Bacillus cereus β-lactamase I

Finally, we aimed to demonstrate that epistasis within the catalytic cycle is found in experimental data. Thus, we sought out to find literature that met the following criteria: (_i_) an explicit statement of the enzyme mechanism, (_ii_) the acquisition of all microscopic rate constants that are constituents of $k_{cat}$ and $K_M$, and (_iii_) the availability of all measurements for the wt, both single mutants, and the double mutant. We found one study which met our requirements–an investigation into K73R and E166D in _B. cereus_ β-lactamase I by Gibson and Waley (1990) [45]. We note that two values ($k_3$ for K73R and $k_3$ for K73R/E166D) were reported as thresholds rather than discrete measurements; we simply used the value provided as the threshold for convenience. Furthermore, this study found contribution of specific epistasis, thus we expected to find deviation from the additivity of sub-state free energies, but specifically aimed to extract additional, previously unexplored, deviation stemming from non-specific epistasis. We were unable to find other studies devoid of specific epistasis (_i.e._, those adhering to the additivity assumption) that also reported microscopic rate constants for all relevant mutants (a prerequisite for our analysis of non-specific epistasis), thus necessitating the use of this study.

To extract epistasis, we compared the predicted values of $k_{cat}$, $K_M$, and catalytic efficiency obtained from the fold-changes in the microscopic rate constants _versus_ the measured kinetic parameters. First, to ensure microscopic rate constants could be accurately used to compute the kinetic parameters, we determined whether the calculated values for the kinetic parameters in each variant matched the measured values. We found that six out of twelve computed parameters showed greater than 1.5-fold error to the measured values, suggesting that kinetic parameter computation from microscopic rate constants already introduces a substantial degree of error that may distort predictions (**Table 3**). Thus, to ensure that the error was systematically represented in calculations of epistasis, we ignored the experimentally measured kinetic parameters and exclusively used the computed kinetic parameters to obtain fold-changes for each mutant (see Methods).

First, as expected, we found significant, specific epistasis between K73R and E166D in _B. cereus_ β-lactamase I. In all cases, the double mutant exhibited diminishing losses in the microscopic rate constants due to positive epistasis: 21.0-fold for $k_1$, 6.3-fold for $k_{-1}$, 16.7-fold for $k_2$, and 13.5-fold for $k_3$. Next, however, we found weak, but significant, non-specific epistasis in $K_M$ and catalytic efficiency, but none in $k_{cat}$ (**Fig 5a**-**5c**). This means that even in the absence of specific epistasis seen in the microscopic rate constants, we would have observed non-specific epistasis leading to 2.2-fold greater $K_M$ and

**Table 3. Errors associated with computation of kinetic parameters from rate constants *versus* the measured kinetic parameters Gibson and Waley (1990).**

| | Parameter | Measured | Computed | Fold error |
|---|---|---|---|---|
| *wt* | $k_{cat}$ (s⁻¹) | 2200 | 1917 | 1.15 |
| | $K_M$ (µM) | 65 | 73 | 1.13 |
| | $k_{cat}/K_M$ (µM⁻¹ s⁻¹) | 34 | 26 | 1.30 |
| *K73R* | $k_{cat}$ (s⁻¹) | 55 | 44 | 1.25 |
| | $K_M$ (µM) | 101 | 50 | 2.01 |
| | $k_{cat}/K_M$ (µM⁻¹ s⁻¹) | 0.55 | 1.00 | 1.59 |
| *E166D* | $k_{cat}$ (s⁻¹) | 0.67 | 1.00 | 1.64 |
| | $K_M$ (µM) | 8.6 | 7.0 | 1.22 |
| | $k_{cat}/K_M$ (µM⁻¹ s⁻¹) | 0.077 | 0.155 | 2.01 |
| *K73R/E166D* | $k_{cat}$ (s⁻¹) | 0.47 | 0.40 | 1.25 |
| | $K_M$ (µM) | 10 | 2 | 4.67 |
| | $k_{cat}/K_M$ (µM⁻¹ s⁻¹) | 0.047 | 0.176 | 3.74 |

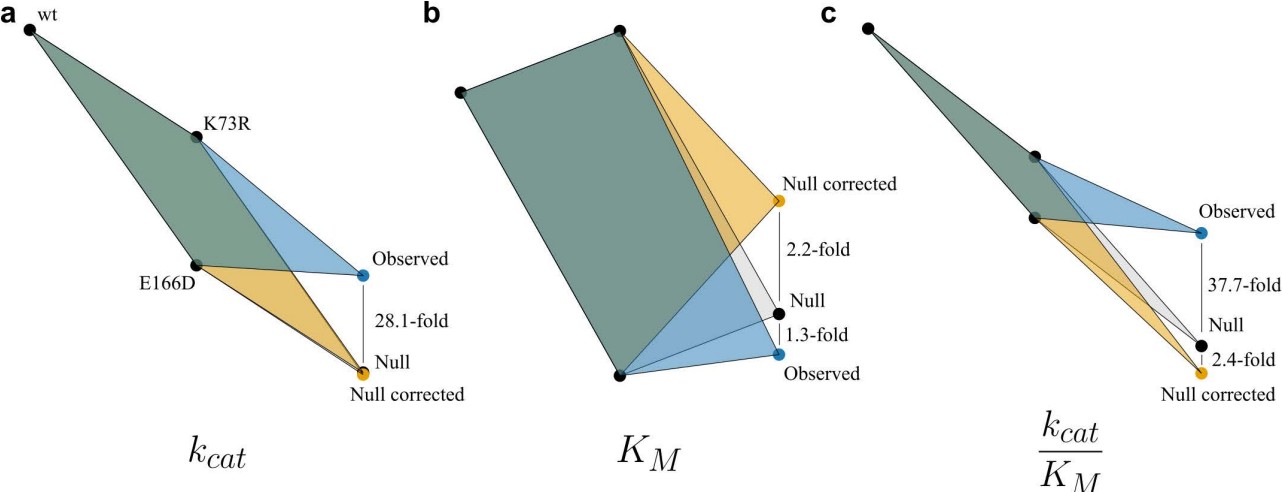

**Fig 5. Correcting for non-specific epistasis in the null model exposes stronger positive effects.** The observed and predicted fold-changes in kinetic parameters of the double mutant based on single mutational effects using a null model and one corrected from non-specific epistasis in the catalytic cycle for **a,** computed $k_{cat}$. **b,** computed $K_M$ and **c,** catalytic efficiency of K73R/E166D in *B. cereus* β-lactamase **I.**

a 2.4-fold lower catalytic efficiency than predicted. In fact, correcting for non-specific epistasis reveals a stronger contribution of specific epistasis than calculated from the null model: the apparently neutral 1.3-fold epistasis in $K_M$ is corrected to 3.0-fold, and the 37.7-fold positive interaction between K73R and E166D in catalytic efficiency results in a corrected 90.6-fold improvement. Although the contribution of non-specific epistasis for these mutations is relatively weak to the effect of specific epistasis, its correction allows for a superior approximation of the greater contribution to diminishing losses in function by the K73R/E166D mutational interaction. Thus, our method is able to extract specific epistasis and, in the case of K73R/E166D *B. cereus* β-lactamase I, it is 2.4-fold stronger in diminishing losses of catalytic efficiency than previously thought.

## Discussion

In this study, we uncovered a novel form of epistasis, intrinsic to the catalytic cycle, that can distort predictions of a mutant's kinetic parameters. By comparing two catalytic cycle models of an enzyme reaction, we demonstrated how increasing complexity in the enzyme mechanism can result in more non-specific epistasis in the double mutants. This form of epistasis can be successfully identified in experimental data, albeit it requires deep functional annotation of the enzyme's microscopic rate constants.

Our findings necessitate a more critical evaluation of epistasis calculated from enzyme kinetic parameters. Threshold models are the contemporary method used for capturing non-specific epistasis; however, they should only be applied when there exists a biochemical basis to assume that the fitted parameter depends on a non-linearity of the underlying additive effects [46]. Aside from rate-limited diffusion and a lower limit of detection defined by the enzyme assay, there is no basis for applying a threshold model to enzyme catalytic parameters. Likewise, the existence of a complex ensemble is often ruled out if enzyme progress curves do not exhibit notable burst or lag kinetics [47]; thus, one may assume that ensemble epistasis is also absent. Hence, many studies have used fold-changes in kinetic parameters of mutants–catalytic efficiency in particular–to connect the changes in structural features of the mutants to the emergent, specific epistasis [11,19,48–52]. Here, we have shown that non-specific epistasis in the catalytic cycle has the potential to distort the contribution of specific epistasis, not just in magnitude but also in sign. Indeed, it is very unlikely that kinetic parameters such as catalytic efficiency and the Michaelis-Menten constant will be free of non-specific epistasis, as that would at the very least require identical modulations of the binding and catalytic transition states in the enzyme mechanism. This form of non-specific epistasis is likely to be accentuated by changes in the rate-limiting steps for the inferred kinetic parameter during evolution, and shifts in these values will reveal strong epistasis–something we have observed experimentally [48]. Finally, it is in fact rare for enzymes to proceed through a simple Michaelis-Menten mechanism, as many enzymes rely on intermediates or multiple products and/or cofactors, further contributing to the non-specific epistasis that arises. Indeed, model mis-specification and ignorance of an enzyme mechanism's complexity has the potential to further distort epistasis inference; measurements supported by an incorrect model can map free energy changes to the incorrect state in the catalytic cycle, or will represent some blend of free energies preventing accurate distinction between specific and non-specific epistasis [53]. Thus, although difficult to detect, we expect that non-specific epistasis in kinetic ensembles is pervasive.

A deep exploration of non-specific epistasis, particularly its prevalence, patterns, and strength across enzymes, requires a rigorous investigation of each enzyme's mechanism and the characterisation of all microscopic rate constants for the single and double mutants of interest. The stringency of these requirements likely accounts for the scarcity of studies that meet the necessary criteria, as well as the general unawareness of the existence of non-specific epistasis in the catalytic cycle. Collecting the necessary data to explore this form of epistasis is experimentally challenging, requiring expertise in enzymological techniques [54–56]. However, it is a necessary step in formulating accurate hypotheses regarding the impact that non-specific epistasis plays in enzyme evolution and design. Furthermore, the acknowledgement of the possibility of non-specific epistasis in enzyme kinetics is essential for efforts aiming to directly connect structural features within enzyme mutants to the emergent epistasis that arises in functional measurements, as non-specific epistasis obscures this relationship. We therefore encourage researchers to either abstain from implicating structural features of mutants in their explanations of the sources of epistasis in kinetic parameters, or to delve into characterising the enzyme mechanism, in order to accurately address the role of the enzyme's structure in the emergent epistasis.

## Methods

### Simple kinetic model simulations

We defined starting free energies ($G$) for each state as follows: E = 0.0 kcal mol$^{-1}$, (E + S)$^{\ddagger}$ = 10.0 kcal mol$^{-1}$, ES = -5.0 kcal mol$^{-1}$, (ES)$^{\ddagger}$ = 11.0 kcal mol$^{-1}$. We calculated microscopic rate constants for the wt using the following equations:

$$k_1 = Ae^{-\frac{G_{(E+S)\ddagger} - G_E}{RT}}$$

(5)

$$k_{-1} = Ae^{-\frac{G_{(E+S)\ddagger} - G_{ES}}{RT}} \tag{6}$$

$$k_2 = Ae^{-\frac{G_{(ES)\ddagger} - G_{ES}}{RT}} \tag{7}$$

Where the pre-exponential factor $A$ was approximated using transition state theory:

$$A = \frac{k_b T}{h} \tag{8}$$

Where $k_b$ is the Boltzmann constant and $h$ is Plank's constant. We note that this approximation permits $k_1$ to approach $6.22 \times 10^{12}$ M$^{-1}$ s$^{-1}$, which exceeds the rate limit of diffusion by several orders of magnitude. However, for simplicity, we did not modify the constant for substrate binding calculation, and we do not expect results to differ greatly with a more accurate calculation of $A$.

We then computed kinetic parameters as follows:

$$k_{cat} = k_2 \tag{9}$$

$$K_D = \frac{k_{-1}}{k_1} \tag{10}$$

$$K_M = \frac{k_{-1} + k_2}{k_1} \tag{11}$$

$$\frac{k_{cat}}{K_M} = \frac{k_1 k_2}{k_{-1} + k_2} \tag{12}$$

We generated 1,000 mutants $G$ values for each parameter by iteratively drawing samples from a uniform distribution bounded between –2 and 2 and summing the wt $G$ with the drawn sample.

Finally, we created $10^6$ "double mutants" by combinatorially combining all mutational effects from every single mutant. For the predicted kinetic parameter of the double mutant, we obtained the product of the wt kinetic parameter, the fold-change of mutant 1, and the fold-change of mutant 2. For the observed or "true" kinetic parameter, we first computed the microscopic rate constant of the double mutant by calculating the sum of free energy changes to each state in the reaction coordinate based on the single mutation effects. We used these rate constants to compute the observed kinetic parameter of the double mutant.

**Computation of mutational effects on epistasis in $K_M$**

In order to compute epistasis in $K_M$ we used the following equation (derived in **S2 File**):

$$\varepsilon_{K_M} = \frac{\alpha_1 \alpha_2 k_{-1} + \beta_1 \beta_2 k_2}{\frac{(\alpha_1 k_{-1} + \beta_1 k_2)(\alpha_2 k_{-1} + \beta_2 k_2)}{k_{-1} + k_2}} \tag{13}$$

We then computed epistasis for combinations of $\alpha_1$, $\alpha_2$, $\beta_1$, and $\beta_2$ ranging from 0.01 to 100. For the initial computation $k_{-1} = 62.1$ s$^{-1}$ and $k_2 = 11.5$ s$^{-1}$. Subsequent tests with variations in $k_{-1}$ were performed at values relative to $k_2$.

## Complex kinetic model simulations

We followed the steps as the more complex model, with additional energy states of $(E+P)^{\ddagger}=9.0$ kcal mol$^{-1}$ and EP=-9.0 kcal mol$^{-1}$ as well as the same definitions for $k_1$, $k_{-1}$, and $k_2$, in addition to the new rate constants:

$$k_{-2} = Ae^{-\frac{G_{(ES)\ddagger} - G_{EP}}{RT}} \tag{14}$$

$$k_3 = Ae^{-\frac{G_{(EP)\ddagger} - G_{EP}}{RT}} \tag{15}$$

And updated kinetic parameters:

$$k_{cat} = \frac{k_2 k_3}{k_2 + k_{-2} + k_3} \tag{16}$$

$$K_M = \frac{k_2 k_3 + k_{-1}k_{-2} + k_{-1}k_3}{k_1(k_2 + k_{-2} + k_3)} \tag{17}$$

$$\frac{k_{cat}}{K_M} = \frac{k_2 k_3 + k_{-1}k_{-2} + k_{-1}k_3}{k_1(k_2 + k_{-2} + k_3)} \tag{18}$$

Where $K_D$ remains the same as Eq. 10.

## Epistasis calculations in B. cerus β-lactamase I

Using equations 4.1, 4.2, and 4.3 we were able to compute $k_{cat}$, $K_M$, and catalytic efficiency using the microscopic rate constants reported in Gibson and Waley (1990) [45]. We found that computed $k_{cat}$ of the single and double mutants was within 1.5-fold error of the measured $k_{cat}$, however the computed $K_M$ for K73R and the double mutant, as well as the computed catalytic efficiency for all mutants, was significantly different (>1.5-fold) than the measured parameters. We opted to use the computed single mutant kinetic parameters outlined in **Table 3**.

Null model fold-changes were calculated using the fold-changes for K73R and E166D which were computed by taking the ratio of the computed mutant parameter and the computed wt parameter. In contrast, corrected null model fold-changes were computed by first calculating the fold-change of each single mutant microscopic rate constant, then computing the expected microscopic rate constant for the double mutant by multiplying the wt constant with the two fold-changes. These expected values were also used for computation of specific epistasis for each microscopic rate constant. Next, we used equations 4.1–4.3 to compute expected kinetic parameters using the expected, not the observed, microscopic rate constants. The difference in these computed kinetic parameters relative to those observed and those computed using the null model provide the different values for epistasis.

## Supporting information

**S1 File. Simple kinetic model simulations.** The Rmarkdown notebook outlining all of the code, with annotations, used to set up the simple kinetic model (outlined in the **Methods** and Fig 1). The document outlines the results and conclusions drawn from the simulation.
(HTML)

**S2 File. Supplementary derivations delineating conditions for non-specific epistasis.** 1) Demonstration of conditions that create non-specific epistasis in the simple kinetic ensemble, 2) Justification for the lack of epistasis in $k_{cat}$ for the simple kinetic model, 3) Justification for the lack of epistasis in $K_D$ for all models.
(DOCX)

**S3 File. Complex kinetic model simulations.** The Rmarkdown notebook outlining all of the code, with annotations, used to set up the more complex kinetic model (outlined in the **Methods** and Fig 4). The document outlines the results and conclusions drawn from the simulation.
(HTML)

## Acknowledgments

We thank Christopher Frøhlich and Adrian H. Bunzel, as well as all members of the Tokuriki lab, for discussions.

## Author contributions

**Conceptualization:** Karol Buda, Nobuhiko Tokuriki.

**Data curation:** Karol Buda.

**Formal analysis:** Karol Buda.

**Funding acquisition:** Nobuhiko Tokuriki.

**Investigation:** Karol Buda, Nobuhiko Tokuriki.

**Methodology:** Karol Buda.

**Project administration:** Nobuhiko Tokuriki.

**Resources:** Karol Buda.

**Software:** Karol Buda.

**Supervision:** Nobuhiko Tokuriki.

**Validation:** Karol Buda.

**Visualization:** Karol Buda.

**Writing – original draft:** Karol Buda.

**Writing – review & editing:** Karol Buda, Nobuhiko Tokuriki.

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
