## [Decision Letter · Decision Letter 0]

1 Dec 2025

Free energy perturbations in enzyme kinetic models reveal cryptic epistasis

PLOS Computational Biology

Dear Dr. Tokuriki,

Thank you for submitting your manuscript to PLOS Computational Biology. After careful consideration, we feel that it has merit but does not fully meet PLOS Computational Biology's publication criteria as it currently stands. Therefore, we invite you to submit a revised version of the manuscript that addresses the points raised during the review process.

We look forward to receiving your revised manuscript.

Kind regards,

Eric C. Dykeman, Ph.D.

Academic Editor

PLOS Computational Biology

Feilim Mac Gabhann

Editor-in-Chief

PLOS Computational Biology

**Additional Editor Comments (if provided):**

In addition to addressing the concerns of Reviewers 2 and 3, can you please provide a detailed response to critiques of Reviewer 1, who seems to suggest there is limited applicability of your model and specifically that the results related to Beta lactamase in B. cereus. are incorrect.

**Journal Requirements:**

**Reviewers' comments:**

Reviewer's Responses to Questions

**Comments to the Authors:**

Reviewer #1: The manuscript examines how non-specific epistasis in enzyme kinetic parameters arises from the nonlinear map that connects

these parameters to the underlying elementary transition rates. The latter are taken to be of Arrhenius form with free energies

that are assumed to change on an additive scale (that is, the free energy change associated with a double mutant is the sum

of the free energy changes of the constituent single mutations). This assumption, which is central to the entire investigation,

seems to me to be completely unfounded, and in fact, the empirical example of the Bacillus cereus beta-lactamase that the authors

discuss shows it to be manifestly wrong.

Although this point is not made explicit in the manuscript, I understand that the authors don't assume mutational additivity of

free energies because this is necessarily realistic, but in order to bring out the non-specific contribution to epistasis, which

requires a (fictional) additive trait from which epistasis is generated through the nonlinear map to the function of interest. But

even if I accept the setting of the study as motivated by the (popular, but, to my mind, not very helpful) distinction between

specific and non-specific epistasis, I feel that the findings are of very limited interest. What is announced somewhat grandly as

"in silico enzyme models" amounts to evaluating the changes in simple rational functions, given explicitly in the Methods section,

under random multiplicative changes of the constituent elementary transition rates. Overall, the work does not substantially advance our understanding of (specific or non-specific) epistasis, and is therefore not suited for publication in PLoS Computational Biology.

Reviewer #2: This is a very valuable contribution in the field of evolutionary genetics, dealing with how non-specific epistasis can arise from even relatively simple enzymatic reaction pathways because individual mutations can affect the energies of multiple reaction steps. Because these energies combine in non-linear ways to determine macroscopic reaction constants, this leads to apparent epistasis, even for mutations in which all effects on the level of energies are strictly additive. This paper builds on the idea of ensemble epistasis but is honest about that precedent. I find this work very relevant, because deep mutational scanning and quantification of epistasis is becoming an ever more widely used technique, especially in enzymes, and it seems very important to me that its practitioners are aware of the sources of epistasis in their datasets.

I have no issues with the analytical results in this study. I only have one question the authors may comment on: In many enzymes the exact reaction pathway may not be known in all its complexity. The authors already show that taking care of the ensemble epistasis in one case can lead to different values for the specific epistasis in the system. How serious a problem is model mis-specification (i.e. a simplified kinetic scheme relative to the more complex biophysical reality) for our desire to quantify specific epistasis?

Another comment would be whether we have much evidence that individual mutations usually affect multiple energy levels along the reaction coordinate. It’s not an unreasonable, assumption, but I am wondering whether the authors can cite more examples where this is known to be true.

Finally, I offer my sincere apologies for how late this review is. I judge this to be an important piece of work so did not want to pass up on the opportunity to review it.

Reviewer #3: In this work, the authors explore the properties of enzyme kinetic models and describe a property that they term cryptic epistasis, meaning that although mutations may appear to have linear effects on basic enzyme properties (free energies), they can combine in non-linear ways to affect observed kinetic properties such as Km and kcat. The paper is clearly written for the most part. However, I hope that the authors can clarify a few uncertainties I had about their findings and methods, especially with respect to the novelty of the described ‘cryptic epistasis’, as at least superficially to me the result is expected from basic enzyme rate law expressions and not surprising in the least. I hope the authors can provide a clarification if I am mistaken in that view. That said, the practical application of their analysis to beta-lactamase double mutants seems valuable and interesting in itself. Specific comments below.

Comments

- Enzyme kinetic parameters (e.g. Km, kcat) are of course well-established to be non-linear combinations of microscopic reaction rate constants (e.g. k1, k2) or reaction free energies(dG or equivalently K), which are presumably what is affected by mutations affecting reaction step free energies. Thus, I am confused why it is surprising that affecting microscopic parameters would have a non-linear effect on macroscopic parameters. Can the authors better justify the current thinking in the field on this claim, with citations?

- I am also unsure whether the authors can definitively state that a mutation would affect a single parameter (i.e. an individual free energy) in a linear/additive way with other mutations (if I understand their methodology correctly – feel free to correct me) – how realistic is this assumption itself? Can the authors assess this from the beta-lactamase case?

- Who says “Enzyme kinetic parameters are assumed to be devoid of non-specific epistasis when measured in vitro” – why would this be the case? What is the basis for this assumption? In the Discussion the authors state that it is commonly believed that “the confounding mutational effects on the physical properties of the enzyme can be controlled through appropriate experimental design,” but I am having trouble even conceptualizing how experimental design of an enzyme assay could control the degree of epistasis of mutations within the enzyme on its observed macroscopic kinetic parameters (Km, kcat).

- In several places it seems to me the authors could better describe specific and non-specific epistasis. In particular, the authors state that non-specific epistasis is when a non-linear mapping is present, but isn’t this the definition of epistasis in general, when two mutations combine in non-linear (i.e. non-additive) ways? Maybe a more precise phrasing would be possible here, to distinguish from epistasis where mutations physically interact, which is a clear concept of course.

**Have the authors made all data and (if applicable) computational code underlying the findings in their manuscript fully available?**

Reviewer #1: Yes

Reviewer #2: Yes

Reviewer #3: Yes

PLOS authors have the option to publish the peer review history of their article (what does this mean? ). If published, this will include your full peer review and any attached files.

**Do you want your identity to be public for this peer review?** For information about this choice, including consent withdrawal, please see our Privacy Policy .

Reviewer #1: No

Reviewer #2: No

Reviewer #3: No

**Figure resubmission:**

**Reproducibility:**



---

## [Decision Letter · Decision Letter 1]

2 Feb 2026

PCOMPBIOL-D-25-01794R1

Free energy perturbations in enzyme kinetic models reveal cryptic epistasis

PLOS Computational Biology

Dear Dr. Tokuriki,

Thank you for submitting your manuscript to PLOS Computational Biology. After careful consideration, we feel that it has merit but does not fully meet PLOS Computational Biology's publication criteria as it currently stands. Therefore, we invite you to submit a revised version of the manuscript that addresses the points raised during the review process.

We look forward to receiving your revised manuscript.

Kind regards,

Eric C. Dykeman, Ph.D.

Academic Editor

PLOS Computational Biology

Feilim Mac Gabhann

Editor-in-Chief

PLOS Computational Biology

**Additional Editor Comments (if provided):**

**Journal Requirements:**

**Reviewers' comments:**

Reviewer's Responses to Questions

**Comments to the Authors:**

Reviewer #1: The authors have provided detailed responses to all my comments. Although I remain unconvinced of the broader significance of the study, my specific points of criticism have been addressed and I do not object to the publication of the manuscript in PloS Comp. Biol. However, I ask the authors to bring out two aspects of their response more clearly in the manuscript text.

1. The empirical evidence for the additivity of transition state free energies (as presented in the new reference by Wells and other work) is a crucial basis for the study. Since it was unknown to me, it may be unknown to other readers as well. It should therefore be mentioned and discussed explicitly early in the introduction.

2. Similarly, the fact that the beta-lactamase system does *not* conform to the additive pattern (but does not disprove the latter, because it is known to be an exception) should be mentioned explicitly either in the Introduction or in the corresponding Results section, and it should be explained why the authors nevertheless choose to discuss this exammple

Reviewer #2: My concerns have been addressed

Reviewer #3: The authors have sufficiently clarified my areas of confusion in the previous draft. I have no additional concerns.

**Have the authors made all data and (if applicable) computational code underlying the findings in their manuscript fully available?**

Reviewer #1: Yes

Reviewer #2: Yes

Reviewer #3: None

PLOS authors have the option to publish the peer review history of their article (what does this mean? ). If published, this will include your full peer review and any attached files.

**Do you want your identity to be public for this peer review?** For information about this choice, including consent withdrawal, please see our Privacy Policy .

Reviewer #1: No

Reviewer #2: No

Reviewer #3: No

**Figure resubmission:**
---

## [Decision Letter · Decision Letter 2]

2 Mar 2026

Dear Dr. Tokuriki,

We are pleased to inform you that your manuscript 'Free energy perturbations in enzyme kinetic models reveal cryptic epistasis' has been provisionally accepted for publication in PLOS Computational Biology.

Best regards,

Eric C. Dykeman, Ph.D.

Academic Editor

PLOS Computational Biology

Feilim Mac Gabhann

Editor-in-Chief

PLOS Computational Biology

Reviewer's Responses to Questions

**Comments to the Authors:**

Reviewer #1: My comments have been fully addressed. The manuscript can be published as is.

**Have the authors made all data and (if applicable) computational code underlying the findings in their manuscript fully available?**

Reviewer #1: Yes

PLOS authors have the option to publish the peer review history of their article (what does this mean? ). If published, this will include your full peer review and any attached files.

**Do you want your identity to be public for this peer review?** For information about this choice, including consent withdrawal, please see our Privacy Policy .

Reviewer #1: No

---

## [Editor Report · Acceptance letter]

PCOMPBIOL-D-25-01794R2

Free energy perturbations in enzyme kinetic models reveal cryptic epistasis

Dear Dr Tokuriki,

I am pleased to inform you that your manuscript has been formally accepted for publication in PLOS Computational Biology. Your manuscript is now with our production department and you will be notified of the publication date in due course.

With kind regards,

Anita Estes
